# REPRESENTING SIGNS AS SIGNS: ONE-SHOT ISLR TO FACILITATE FUNCTIONAL SIGN LANGUAGE TECHNOLOGIES

## ABSTRACT

Isolated Sign Language Recognition (ISLR) is crucial for scalable sign language technology, yet language-specific approaches limit current models. To address this, we propose a one-shot learning approach that generalises across languages and evolving vocabularies. Our method involves pretraining a model to embed signs based on essential features and using a dense vector search for rapid, accurate recognition of unseen signs. We achieve state-of-the-art results, including 50.8% one-shot MRR on a large dictionary containing 10,235 unique signs from a different language than the training set. Our approach is robust across languages and support sets, offering a scalable, adaptable solution for ISLR. Co-created with the Deaf and Hard of Hearing (DHH) community, this method aligns with real-world needs, and advances scalable sign language recognition.

## 1 INTRODUCTION

Isolated sign language recognition (ISLR) is a crucial first step toward achieving full sign language translation (SLT). Despite notable progress, most advances in ISLR have been confined to specific languages and respective datasets, limiting the range of vocabularies that can be recognised in real-world applications. In addition, with sign languages constantly evolving, this limitation becomes increasingly problematic. More flexible approaches are required – ones that allow for the expansion of vocabularies and are robust enough to handle large-scale dictionaries for SLT.

To overcome these challenges, it is essential to shift the focus away from predefined vocabularies and instead emphasise the intrinsic features of signs. Once an *effective* representation of a specific sign is established, the surrounding context can be integrated afterwards. An *effective* sign representation requires two key elements: (1) each sign is uniquely identifiable, and (2) only the essential features, i.e. sign phonemes, are captured to ensure reliable discrimination and adaptability across varied signing conditions.

Following this idea, we propose leveraging one-shot learning for sign language recognition. One-shot learning is particularly well-suited for sign language recognition since it allows the model to generalise from limited examples, making it possible to recognise new signs without extensive retraining or data collection. By focusing on embedding signs efficiently, this method enables accurate, context- and even language-independent isolated sign recognition.

Our approach consists of two steps: (1) pretraining a model that can reliably embed signs, and (2) using this frozen pretrained model for a dense vector search. In the first step, we train the model on a diverse set of signs to capture their essential features. This creates embeddings representing signs as points in a high-dimensional space. In the second step, a dense vector search matches new, unseen variations of signs to the closest embedding in this space, enabling rapid and accurate recognition without requiring further retraining.

We evaluate both the pretraining and one-shot classification, achieving state-of-the-art results. Notably, we attain a 50.8% one-shot MRR on an extremely large dictionary from a language entirely different from the pretraining data. Our method also demonstrates strong robustness across multiple languages, even when introducing variations in the set of exemplary signs. These findings underscore that ISLR systems can be generalised across languages, independent of the pretraining

data. This enables recognition of vast vocabularies while adapting to the evolving nature of sign languages. This adaptability positions our approach as a progressive solution for enhancing the scalability and effectiveness of sign language technology in diverse contexts.

The development of this paper was guided by a clear message from the DHH community: sign language technology can be initially suboptimal but should be improvable through active feedback, to provide more usable tools for the DHH community *in the short term*. This goal was emphasised at a workshop on sign language research in AI (Bragg et al., 2019). Bearing this in mind, the method of this paper was developed in a co-creation strategy, where the needs and desires of the DHH community were assessed frequently. Ultimately, this led to an openly available tool[1].

**The contributions of this paper are the following. (a)** This paper presents the first robust approach to one-shot sign language recognition, demonstrating its language-independent capabilities and its applicability to larger vocabularies. **(b)** We achieve state-of-the-art results in sign language recognition. **(c)** Our method demonstrates strong generalisation across multiple languages and proves robust even when variations are introduced in the exemplary signs. **(d)** We developed an application using a co-creation strategy in close collaboration with the DHH community, ensuring that sign language research output meets their needs and can be continuously improved through active feedback.

## 2 RELATED WORK

### 2.1 ISOLATED SIGN LANGUAGE RECOGNITION

ISLR is a classification problem. Traditionally, a system analyses a video depicting an isolated sign and aims to predict the corresponding label or gloss. These input videos can be processed in several ways. Recent advancements (Papadimitriou & Potamianos, 2023; Chen et al., 2022) have demonstrated a positive impact using pose estimation models, such as MediaPipe Holistic (Grishchenko & Bazarevsky, 2020) (henceforth MediaPipe) and OpenPose (Cao et al., 2017). These models transform input videos into sequences of skeletal representations, capturing "keypoints" or "landmarks" of the human pose in 2D or 3D Cartesian coordinates. By removing all information about a person's appearance, these tools enhance the generalisability to downstream tasks. This transformation allows ISLR models to focus solely on the structural aspect of sign videos: how people move their arms, hands, and face.

MediaPipe has significantly advanced the current state-of-the-art of ISLR, but there is still considerable room for improvement. Moryossef et al. (2021) argued that this tool is not directly applicable to fine-grained tasks like sign language recognition. Although the keypoint estimator is generally accurate, MediaPipe struggles when two body parts interact. Since this interaction is elemental to sign language, crucial information is lost. However, recent Kaggle competitions (Chow et al., 2023b;a) based on keypoint estimation using MediaPipe present a different perspective, showing promising results in SLR using keypoint estimation. A key component appears to be the addition of a frame embedding that is not present in the work by Moryossef et al., but present in all top Kaggle competition entries. This frame embedding allows the network to learn the non-linear relationships between keypoints (De Coster et al., 2023).

Not only the preprocessing of sign language videos is essential. The architecture of the models also has a great impact. Until 2020, deep learning approaches to ISLR primarily used variations of Recurrent Neural Networks (Koller et al., 2016; 2017; Ye et al., 2018). The common factor of these models is that they are proficient at handling sequential data and dealing with temporal dependencies between different poses. The introduction of transformers (Vaswani et al., 2017) in 2017 initiated a paradigm shift. The combination of keypoints and attention leads to powerful models for ISLR: the top scoring on the Kaggle ASL ISLR competition (Chow et al., 2023b) method achieved 89.3% test set accuracy on 250 sign categories using keypoint data.

### 2.2 FEW-SHOT SIGN LANGUAGE RECOGNITION

The evolving nature of sign languages motivates the need for flexible ISLR techniques. The traditional sign language classification models described in the previous section lack this flexibility, as

---

[1]Placeholder for the link to the online tool, currently not included due to double-blind review policy.

they are limited to the glossary provided in the training set and require a large number of examples for every sign. Fei-Fei et al. (2006) argued that "one can take advantage of knowledge coming from previously learned categories, no matter how different these categories might be." This insight led to the introduction of few-shot learning, where fewer examples per category are required.

There are various gradations to few-shot learning, including zero-shot learning. In zero-shot learning, the model has never seen an example for a given category and relies on its prior knowledge and some form of description of the category. Zero-shot learning was first introduced to SLR by Bilge et al. (2019). Their method involved matching textual descriptions to video inputs, utilising BERT text embeddings (Devlin et al., 2018) and 3D CNNs alongside bidirectional LSTMs for video processing. More recently, Rastgoo et al. (2021) approached zero-shot ISLR with a similar technique. This method combined BERT text embeddings with pose estimators and vision transformers, followed by LSTMs for temporal modelling. These zero-shot methods rely on the presence of a textual description of a sign, which is not always available.

More often, there are one or multiple example videos available for one sign, which can be used for few-shot learning. If none are available for a given sign, recording one example is easier than accurately describing the sign through text. Wang et al. (2021), for example, leverage multiple examples (i.e., they perform few-shot learning) of one sign to perform K-means clustering and a custom matching algorithm. For some sign languages, a dictionary is available. In the case of VGT, the dictionary contains exactly one example per unique sign, which is ideal for one-shot learning. De Coster & Dambre (2023) in essence performed one-shot learning to recognise signs in the VGT dictionary (Van Herreweghe et al., 2004). In their work, a pretrained model outputs embeddings, which are used in a Euclidean distance-based vector search. However, despite being pretrained on VGT data, the model's performance on dictionary lookup was suboptimal. This shortfall can be attributed to the Zipfian class distribution in the dataset and the fact that the examples come from continuous signing rather than isolated signs.

## 3 METHODOLOGY

### 3.1 ONE-SHOT SIGN LANGUAGE RECOGNITION

Our one-shot inference exists out of two steps, namely: initialisation and inference. These steps are illustrated in fig. 1. In the initialisation step – illustrated with solid lines – the frozen model converts all dictionary videos to embeddings. This collection of embeddings is known as the support set, which serves as a reference for subsequent comparisons during the inference phase. During the inference step – depicted with dashed lines – the model processes an input query video and generates its corresponding embedding. This embedding is then compared against the support set to determine the most similar entry. The comparison utilises the attention mechanism as described by (Bahdanau et al., 2014), which effectively identifies the closest match from the support set solely based on the embeddings.

In essence, this method performs *search* as described by Mittal et al. (2021), who formulated it as the "selection of a relevant entity from a set via *query-key* interactions". In our case, the queries and keys are the query videos and the support set respectively. Furthermore, this approach applies the softmax function, and hence the output can be interpreted as probabilities. These probabilities provide a measure of confidence in the match, which in turn allows for more nuanced decision-making, ultimately improving the system's interpretability and reliability.

We evaluate the one-shot classification method using two approaches. The first evaluation involves a limited query set with a large support set, assessing the model's ability to handle applications requiring extensive vocabularies, such as SLT. The second experiment introduces variability by randomly selecting different instances from a sign language dataset multiple times, effectively reconstructing the same classes within the support set. By measuring the spread of evaluation metrics across these variations, we investigate how the variability within sign categories and the selection of support set instances influence our approach.

Although this one-shot technique is remarkably elegant, the performance heavily relies on how the manual sign features are represented within the embedding. Two main factors independently impact this objective: the pretraining and the utilised datasets. We employ isolated sign language

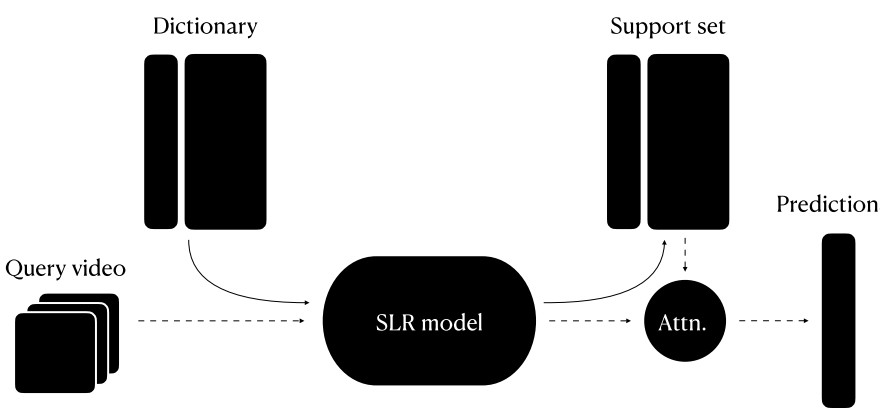

Figure 1: We perform one-shot sign classification to search through a dictionary with a query video. Solid arrows: a sign language dictionary is mapped to a support set of embeddings by an SLR model. This is done once. Dashed arrows: we can classify a new example (query the dictionary) by also mapping the example to an embedding with the same model and using attention to obtain probabilities for every label in the support set. This can be done without regenerating the support set.

recognition for pretraining since it aligns best with our downstream goal. Furthermore, both the quality and quantity of the used pretraining data significantly impact the results of the downstream tasks, as the results in section 4.2 show.

## 3.2 DATASETS

Several kinds of datasets are required to perform our experiments. First, we need one or more pretraining datasets, on which we can train a model for ISLR. Second, datasets for the one-shot ISLR evaluation are needed. These datasets include both real-world sign language dictionaries and existing isolated sign language datasets that we have adapted for the one-shot task.

### 3.2.1 PRETRAINING DATASETS

One primary example of one-shot ISLR is dictionary retrieval, like the study by De Coster & Dambre (2023). To facilitate comparisons with this work, the same pretraining dataset is utilised. This dataset is derived from the VGT corpus (Van Herreweghe et al., 2015). However, the limited number of classes and the severe class imbalance in this dataset may have contributed to suboptimal performance. Therefore, we look for a larger and more varied dataset for ISLR: we choose ASL Citizen (Desai et al., 2024) for its sizeable vocabulary and because DHH signers recorded the signs.

The VGT dataset consists of 24,967 examples for 292 signs, but these examples are not uniformly distributed (fig. 2a). Since the dataset is derived from a corpus that contains spontaneous language use, the distribution is rather Zipfian. The majority of examples represent only a minority of the signs. The examples are cut from continuous signing, which means that every sign execution is influenced by preceding and subsequent signs, a phenomenon referred to as co-articulation.

The ASL Citizen dataset is richer, containing 83,399 examples for 2,731 signs. These examples are more uniformly distributed, with an average of 30 instances per sign (fig. 2b). Since the dataset entries were recorded in isolation, no co-articulation occurred. This more closely matches the task of dictionary search. In fact, the ASL Citizen dataset was envisioned as a dataset of "in-the-wild dictionary queries" (Desai et al., 2024).

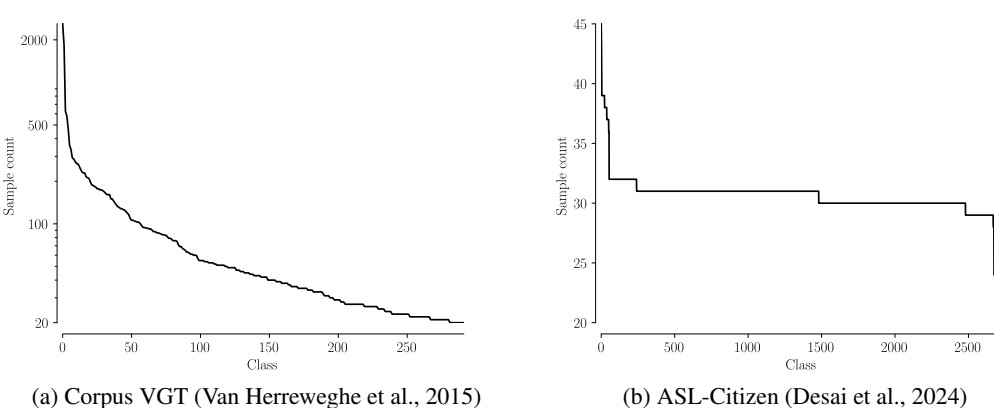

(a) Corpus VGT (Van Herreweghe et al., 2015)    (b) ASL-Citizen (Desai et al., 2024)

Figure 2: Class distributions of pretraining datasets, sorted by descending sample count.

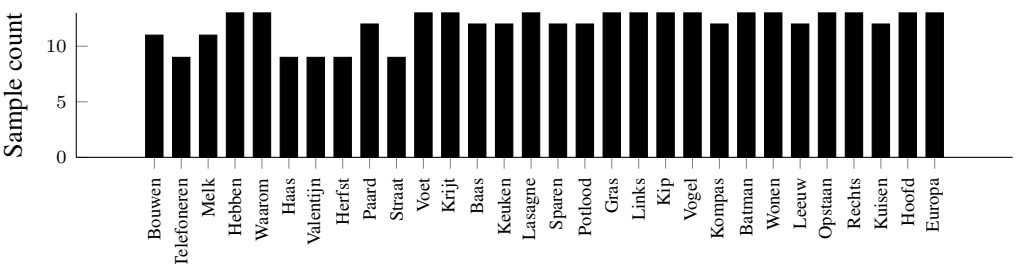

Figure 3: The number of dictionary queries per gloss is distributed approximately uniformly with mean 11.93. Each label on the horizontal axis is a link to the corresponding video within the Flemish Sign Language dictionary.

### 3.2.2 EVALUATION DATASETS

One application of our work is enabling native and non-native signers to search through sign language dictionaries by recording or uploading a video of a sign. To evaluate the performance of a dictionary search system (one-shot classifier) we always require two sets: dictionary entries, referred to as the support set, and dictionary queries, which serve as test examples. Both sets consist of short videos containing individual signs that mirror the entries found in the dictionary. Ideally, these signs are performed in citation form, meaning that each sign starts and ends in a resting position.

Due to sign language dialects and synonyms, multiple signs may correspond to a single word or concept. However, we do not consider this as the same sign and instead label signs by unique IDs. In this work, we focus on retrieving the exact sign matching the query video, rather than its meaning in spoken language.

For the first evaluation set, we collect a set of dictionary queries in VGT for 30 distinct sign categories (fig. 3)[2]. The 10 categories used in prior dictionary retrieval research (De Coster & Dambre, 2023) are a subset of our set of 30. As the first support set, we use a pre-existing sign language dictionary, that is, the VGT dictionary (Van Herreweghe et al., 2004).[3] This dictionary contains 10,235 unique signs at the time of writing. Like many other sign language dictionaries, every sign has exactly one example, which is why we opt for one-shot ISLR.

The second set of evaluation datasets includes AUTSL (Sincan & Keles, 2020) and WLASL (Li et al., 2020), which consist of Turkish and American Sign Language respectively. The WLASL dataset provides splits for various dictionary sizes – 100, 300, 1000, and 2000 signs – offering a

---

[2]This dataset will be made publicly available after publication.

[3]The dictionary videos can be downloaded from this website: `https://taalmaterialen.ivdnt.org/download/woordenboek-vgt/`.

robust evaluation for scaling dictionary sizes. AUTSL, on the other hand, comprises a vocabulary of 226 independent signs. For all experiments using these datasets, the test set serves as queries, while the support set is constructed by randomly selecting one entry per class from the training set for 100 different times. To ensure these sets were different, seeds were used. Random sampling provides a more rigorous assessment of the model's performance, demonstrating that its performance does not depend on the quality or consistency of individual dictionary entries.

### 3.3 Pretraining: Keypoint-based Sign Language Recognition

To perform one-shot sign language recognition, we first need to pretrain a sign language recognition model. We choose to use a keypoint-based model. By performing human keypoint estimation on the sign language videos, the sign language recognition task is facilitated. Moreover, we hypothesise that using a keypoint-based model is beneficial to the one-shot classification task, because it reduces the impact of the visual properties (background, lighting, clothing, etc.) of the query and the dictionary videos, and it better aligns the input distributions of the data used in the pretraining step and the querying step. Another advantage of using keypoints is the ability to integrate publicly available estimators, such as MediaPipe, into client-side applications, ensuring both privacy and low-latency communication.

More specifically, we choose to employ MediaPipe (Grishchenko & Bazarevsky, 2020) for the reasons listed in section 2.1. MediaPipe predicts the pose, hands and face keypoints. We only utilise the pose and hand information. Thus, our approach solely focuses on the manual components of signs: handshape, movement, place of articulation, and orientation. These manual features are transferable across sign languages. We also trained, optimised and tested embedding models that use mouth keypoints, but the results for one-shot recovery were worse. This decrease can be attributed to the fact that mouthings are more language-specific (Bank et al., 2015). Therefore, we do not deem them essential to this work, but acknowledge their importance in broader sign language processing.

We build on the architectural style of the SignON research project (Holmes et al., 2023), making slight modifications to the number of layers and blocks to better fit the requirements of our task. A schematic overview of this so-called PoseFormer network is shown in fig. 4. The network integrates dense and convolutional blocks, followed by a multi-head attention mechanism.

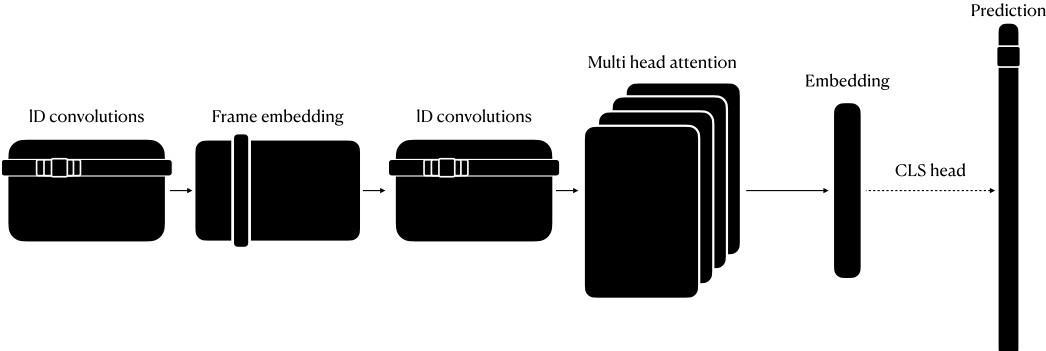

Figure 4: The PoseFormer model, represented by solid lines, consists of several blocks. The 1D convolutions process data along the temporal axis, while the frame embedding block handles individual frames. Finally, the multi-head attention block extracts relevant features. After training, the classification head, consisting of a single linear layer (depicted with a dashed line), is removed.

In this model, a sequence of keypoint coordinates representing human poses serves as input. Initially, 1D convolutions are used for temporal smoothing, capturing short-term dependencies in the sequence. The output is then fed into a multi-layer dense sub-network, which processes each frame individually to extract non-linear representations of the pose. These frame-level representations are subsequently passed through a convolutional block that focuses on learning local temporal context. Finally, a multi-head self-attention mechanism captures global temporal dependencies across the sequence. The resulting vector, representing each sign, is input into a linear classifier for final prediction.

Table 1: We optimise hyperparameters for both models

| Hyperparameter | ASL | VGT |
|---|---|---|
| Batch size | 64 | 128 |
| Learning rate | 0.0003 | 0.0003 |
| Representation size | 160 | 192 |
| Attention layers | 4 | 4 |
| Attention heads | 8 | 8 |
| Dropout | 0.2 | 0.2 |

As noted in section 3.2.1, we use two pretraining datasets with different properties and compare results obtained with them. When we pretrain the PoseFormer with the VGT dataset, we refer to this model as PF-VGT, and when we pretrain it with the ASL Citizen dataset, we refer to it as PF-ASL. The used hyperparameters for both datasets are represented in table 1.

## 3.4 EVALUATION METRICS

Several evaluation metrics are used throughout the entire pipeline to ensure a robust evaluation of the model. First, we report the mean Recall@$K$ for $K \in [1, 5, 10]$. Furthermore, two ranking metrics are employed: mean reciprocal rank (MRR) and normalised discounted cumulative gain (nDCG). Given that there is only one relevant item per prediction (its label), the interpretations of MRR and nDCG are similar in this scenario. The key difference is that the inverse of the MRR is the harmonic mean of the ranks of all predictions, which provides an alternative view of the results. For all three metrics, higher is better and a value of one indicates optimal performance. All three metrics – Recall@$K$, MRR, and nDCG – are used to evaluate the pretraining, to allow for comparison with a baseline ISLR model. For the evaluation of the one-shot methods, reporting only the Recall@$K$ and the MRR is deemed sufficient, since the interpretation of MRR and nDCG is very similar.

## 4 RESULTS

### 4.1 PRETRAINING

The pretraining stage results are summarised in table 2. The I3D model (Desai et al., 2024) is considered the baseline: it is, as of writing, the best-performing model in the scientific literature for ASL Citizen. We compare the PoseFormer model with this baseline. To further assess the PoseFormer's architecture, we conduct a limited ablation study, isolating the impact of its key components. Specifically, these components are: the input convolutions and intermediate convolutions, which appear respectively before and after the frame embedding sub-network. We remove these components individually and measure the impact with respect to our metrics.

Table 2: The PoseFormer outperforms the I3D baseline on the pretraining task (ASL Citizen). The ablation study illustrates the importance of the frame embedding and convolutions in the Pose-Former.

| Model | ↑ MRR | ↑ nDCG | ↑ Rec@1 | ↑ Rec@5 | ↑ Rec@10 |
|---|---|---|---|---|---|
| Poseformer | **0.833** | **0.870** | **0.751** | **0.932** | 0.955 |
|   No input conv. | 0.831 | 0.869 | 0.749 | 0.931 | **0.956** |
|   No frame embedding | 0.811 | 0.854 | 0.723 | 0.920 | 0.946 |
|   No intermediate conv. | 0.819 | 0.860 | 0.733 | 0.926 | 0.951 |
| I3D (Desai et al., 2024) | 0.733 | 0.791 | 0.631 | 0.861 | 0.909 |

Table 2 illustrates the PoseFormer's significant improvement over the I3D baseline (Desai et al., 2024) (+ 0.0993 MRR and + 0.1199 Recall@1). As of writing, our results are the state of the art on the ASL Citizen dataset. The ablation study emphasises the importance of the components of the PoseFormer. The input convolutions, which act as temporal filters on the raw keypoint coordinate features, have limited impact on the scores. The frame embedding is also present in the original network, and has a larger impact (+ 0.0211 MRR and + 0.0278 Recall@1). The intermediate convo-

lutions, which appear in the network between the frame embedding and self-attention blocks, also have a larger impact than the input convolutions (+ 0.0132 MRR and + 0.018 Recall@1). These results confirm the relevance of the individual components of the PoseFormer Network.

## 4.2 ONE-SHOT SIGN LANGUAGE RECOGNITION

There are two sets of results for the one-shot ISLR. The first considers the PF-ASLmodel's capabilities to handle extremely large dictionaries. The second set evaluates both models' resilience to the selection of support set samples. For this set of experiments, we only consider the Recall@$K$ and the MRR, since the interpretation of MRR and nDCG is very similar.

### 4.2.1 LARGE DICTIONARIES

Figure 5a displays the relation between the dictionary size and Recall@$K$. We refer to the results of the independent models using the names from section 3.3: PF-VGT and PF-ASL (i.e., the Pose-Former pretrained on VGT and ASL respectively). For a dictionary of 100 signs, PF-ASL has a Recall@1 of 0.888. PF-VGT, despite being pretrained on the language of this dictionary lookup task, achieves a score of only 0.506. For the full dictionary of 10,235 signs, PF-ASL achieves a Recall@1 score of 0.374, which is a substantial improvement over the 0.089 score obtained by PF-VGT. To put these results into perspective, a random search through the 10,235 classes would yield a Recall@1 of only 9.77e-5. These results underscore the impressive performance of PF-ASL in accurately predicting the correct class from a vast dictionary.

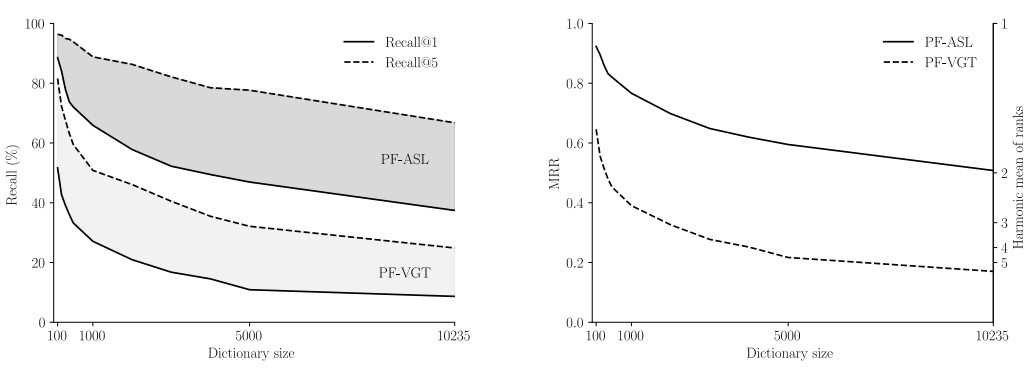

(a) Recall@1 and Recall@5 for large dictionary search       (b) MRR for large dictionary search

Figure 5: Metrics for one-shot evaluation

For the PF-ASL, although not optimal, the MRR (fig. 5b) remains relatively high at 0.508 across 10,235 categories. This suggests that, on average, the correct prediction ranks second. On the other hand, Recall@5 offers a complementary perspective: for the same 10,235 categories, the correct result is among the top five predictions 67% of the time. This demonstrates the model's ability to consistently rank the correct prediction within a small set of top choices.

### 4.2.2 SUPPORT SET PERTURBATION

The results on the support set perturbation are given in table 3. We only report the Recall@1 and MRR for both the PF-VGT and PF-ASL. Once again, these results highlight that the PF-ASL is a robuster model than the PF-VGT, because the pretraining dataset better aligns with the downstream task and contains more different signs. On the AUTSL dataset, we achieved a Recall@1 of 58.1%, while the PF-VGT only achieved a score as high as 40.1%. Although the results differ by almost 20%, the spread on both results is about 3%. The same can be observed in the MRR.

Secondly, we evaluated the WLASL dataset across all predefined dictionary sizes. As expected, test performance diminished with increasing dictionary sizes, with Recall@1 dropping from 0.611 to 0.391. On the largest dictionary size, we also achieved an MRR of 0.500. This means that, on

Table 3: To account for the evolving nature of sign languages, the PoseFormers were tested across various languages. The table also presents the standard deviation associated with perturbations in the dictionary.

| | PF-ASL | | PF-VGT | |
|---|---|---|---|---|
| **Dataset** | **↑ Recall@1** | **↑ MRR** | **↑ Recall@1** | **↑ MRR** |
| WLASL100 | $0.614 \pm 0.024$ | $0.692 \pm 0.020$ | $0.384 \pm 0.017$ | $0.485 \pm 0.017$ |
| WLASL300 | $0.558 \pm 0.011$ | $0.646 \pm 0.010$ | $0.298 \pm 0.008$ | $0.397 \pm 0.009$ |
| WLASL1000 | $0.470 \pm 0.006$ | $0.569 \pm 0.005$ | $0.225 \pm 0.003$ | $0.308 \pm 0.004$ |
| WLASL2000 | $0.391 \pm 0.004$ | $0.500 \pm 0.004$ | $0.198 \pm 0.002$ | $0.269 \pm 0.003$ |
| AUTSL | $0.581 \pm 0.036$ | $0.687 \pm 0.032$ | $0.401 \pm 0.034$ | $0.533 \pm 0.033$ |

average, the correct prediction ranks second. In comparison, PF-VGT places the correct prediction at rank 4.

The results also show a consistently small standard deviation across both datasets, models, and all dictionary sizes, indicating stable performance regardless of dataset variability. This consistency suggests that one-shot ISLR classification is robust to differences in data, maintaining reliable performance across various scenarios. Interestingly, the standard deviation decreases as dictionary size increases, likely due to the larger evaluation set offering a more comprehensive assessment of model accuracy. These findings reinforce the robustness of the models and their strong generalisation capabilities, making them well-suited for real-world applications across diverse settings.

## 5 DISCUSSION

The results reveal four key insights. First, we confirm that keypoint-based models can achieve state-of-the-art results on the challenging isolated sign language recognition task. By utilising keypoints, the one-shot classification task becomes more feasible, as the pose estimator eliminates visual differences between datasets. Second, we demonstrate that the size, vocabulary, and class distribution of the dataset are critical for the pretraining phase, significantly influencing downstream performance in one-shot classification. Next, we find that alignment between the pretraining and downstream languages is less important than these dataset characteristics. Finally, our results highlight that representing individual signs, rather than relying on translations, is feasible and also crucial for creating future-proof sign language technologies. We detail these insights below.

Indeed, the keypoint-based PoseFormer achieves state-of-the-art results on the challenging large-vocabulary ASL Citizen dataset. It outperforms the I3D baseline by 0.120 Recall@1 (a 19% increase). Moreover, the model also transfers seamlessly to downstream tasks on different languages, such as vector-based dictionary search for VGT, as our results illustrate.

The model's performance in the one-shot setting depends on the richness of the pretraining data. Two specific traits of the ASL Citizen dataset enable the high performance of our one-shot classification approach. First, it encompasses a broad vocabulary of 2,731 unique glosses. This large and varied gloss set ensures exposure to a diverse range of signs, which is critical for robust model performance. Second, the ASL Citizen dataset maintains a uniform class distribution, offering sufficient examples across different handshapes and movements. In contrast, the VGT Corpus (Van Herreweghe et al., 2015), used in prior one-shot research (De Coster & Dambre, 2023), follows a Zipfian distribution, where most samples feature the simple pointing handshape. As a result, models trained on VGT struggle to generalise to unseen signs that involve more complex handshapes, as the dataset lacks the necessary variety. This difference in data diversity explains the significantly higher performance of our PF-ASL model compared to PF-VGT. The PoseFormer model, trained on ASL Citizen, benefits from encountering a wide array of handshapes, enabling it to learn more transferable representations for unknown signs.

The variety of the ASL Citizen dataset plays a crucial role in the success of our model in one-shot classification. By ensuring exposure to a wide range of signs, the model minimises the chance of encountering unfamiliar signs, leading to better generalisation. Importantly, our results demonstrate that the diversity within a dataset is more critical than consistency in the language used during pretraining. Even though WLASL (Li et al., 2020) and ASL Citizen (Desai et al., 2024) both represent

ASL, their partially shared, but differing vocabularies yield markedly different results in pretraining and one-shot classification. This suggests that limiting pretraining to a single dataset or language may not capture the full complexity of an entire sign language, and future models should prioritise dataset variety to enhance robustness in real-world applications.

Sign representation plays a pivotal role in achieving robust performance in one-shot sign language classification. By focusing on pose-based embeddings, our approach abstracts away from surface-level visual differences, emphasising core elements of sign structure like handshapes, movement, and orientation. This abstraction enables models to generalise across different languages and datasets, which addresses the challenges posed by variations in signer appearance, environment, and video quality. As sign languages evolve and new signs emerge, systems that rely on such representations rather than static translations are better equipped to adapt and remain relevant, paving the way for scalable and inclusive sign language technologies.

## 6 FUTURE WORK

Despite our considerable gains in ISLR performance and the first results of one-shot ISLR, there are several promising directions for future research. One such direction involves leveraging the lexical information from the ASL-LEX database (Caselli et al., 2017) when using the ASL-citizen dataset. Additionally, extending the existing data with resources like the Sem-Lex benchmark dataset (Kezar et al., 2023) could further enhance recognition performance. However, we argue that the key to achieving more robust sign recognition lies not just in more data but in the diversity of signs it contains. Therefore, multilingual training, where a shared sign representation is used to recognise signs across different languages, may be a more effective approach. Such an approach could vastly expand the potentially recognisable glossary of signs, contributing to more versatile and scalable models.

Finally, the proposed method not only enables highly accurate dictionary search applications but also creates opportunities for other downstream tasks. As mentioned in section 3.1, Mittal et al. (2021) formally described *search*, but also introduced *retrieval* as the extraction of relevant features. We envision the combination of the *search* technique proposed in this paper for the *retrieval* of token embeddings for large language models. The integration into LLMs could facilitate a sign language translation tool or even a sign language-enabled virtual assistant. In summary, due to its effectiveness and lower data requirements, this technique has the potential to significantly advance sign language research, which is currently constrained by data availability. This could lead to the development of more practical tools for the DHH community in the near future.

## 7 CONCLUSION

Sign language recognition models based on keypoints and self-attention, trained on large-vocabulary datasets, can classify unknown signs in a different language with just one training example. We leverage the proven PoseFormer model, pretrain it on the ASL Citizen dataset, and use it in a one-shot classification setting by leveraging the attention mechanism in the embedding space of the internal representations learnt by the PoseFormer. The PoseFormer achieves state-of-the-art sign language recognition on the ASL Citizen dataset (a 19% increase in Recall@1 compared to previous work) and on one-shot sign classification (0.508 MRR and 0.374 Recall@1 on 10,235 signs). For the first time, large vocabulary ISLR is enabled thanks to the one-shot classification approach. Furthermore, we prove that the method generalises to different languages and is independent of the used sign-variations inside the support set. Despite the multi-lingual evaluation, we leave the multi-lingual pretraining for one-shot ISLR for future research. Finally, the results of this paper led to the development of a publicly available dictionary look-up application for the DHH community.

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
