# OpenReview forum: "Representing Signs as Signs: One-Shot ISLR to Facilitate Functional Sign Language Technologies"
_ICLR.cc/2025/Conference — ICLR 2025 Conference Withdrawn Submission_

### Official Review · Reviewer_Stod · 2024-10-29

**Soundness:** 2
**Presentation:** 2
**Contribution:** 2
**Rating:** 3
**Confidence:** 5

**Summary:**

This paper presents a method for one-shot sign language recognition by leveraging a supervised pretrained model to embed signs into a support set. This model enables robust generalization across multiple languages in downstream one-shot recognition tasks. The authors use a keypoint-based model, PoseFormer, to address the dynamic nature of sign languages, achieving state-of-the-art performance on the ASL-Citizen dataset. Additionally, they introduce a new dataset containing dictionary query samples derived from the VGT corpus.

**Strengths:**

(1) **Collaborative Development**: The method was developed through a co-creation strategy with DHH communities to ensure alignment with the community’s needs and priorities.

(2) **Suitability for Evolving Sign Languages**: The proposed method addresses the dynamic and evolving nature of sign languages, demonstrating strong adaptability through one-shot recognition.

(3) **SotA Performance**: The model achieves SotA results on the ASL-Citizen dataset and discuss the effectiveness of keypoint-based models for  isolated sign language recognition.

(4) **Significance of Pretraining Data**: The paper underscores the importance of pretraining data, emphasing the difference between pretraining on ASL-Citizen and VGT.

**Weaknesses:**

(1) **Limited Novelty of Approach**: The novelty of the proposed approach appears somewhat limited. The method essentially trains an existing supervised model on either ASL-Citizen or VGT datasets and then performs a similarity-based lookup for one-shot recognition.

(2) **Limited Technical Details**: The paper lacks sufficient technical information on the training process, aside from the summary in Table 1. Key training details such as the number of frames input into the model, any keypoint augmentations, and the number of epochs used are missing.

(3) **Inadequate Figure Quality and Clarity**: The figures are underdeveloped, primarily showing black-box diagrams that make interpretation challenging. For example, the function of “Attn.” in the figures is unclear, and the text lacks sufficient detail to explain it. Adding equations with labeled symbols to represent the model’s full process could significantly improve readability and understanding.

(4) **SoTA Limited to ASL-Citizen**: While the authors claim SotA results for sign recognition in their contributions, this is only demonstrated on ASL-Citizen, which currently includes only baseline models. The paper should include results on the other datasets used in the paper such as WLASL or AUTSL, to validate their contribution.

(5) **Incomplete Evaluation on the VGT Dataset**: Although the authors mention using the VGT dataset for pretraining and introduce a new derived evaluation set, there are no reported performance results on this dataset and how it compares to the baseline I3D model.

(6) **Insufficient Detail on Co-Creation Strategy**: One stated contribution is the co-creation strategy with DHH communities, but the paper provides minimal detail on this aspect. Expanding on the co-creation process and the specific ways DHH insights shaped the method would add value to this contribution.

**Questions:**

Please see weakness for questions and suggestions.

---

> ### Author Response · Authors · 2024-11-26
>
> We appreciate your feedback. As you point out, our method is simple. We believe its simplicity is its strength, particularly when combined with appropriate pre-training. While ICLR may not be the ideal venue for our work, we will consider submitting a revised version to a more suitable application-oriented conference, incorporating additional evaluations on WLASL and AUTSL.

---

### Official Review · Reviewer_z2LL · 2024-10-31

**Soundness:** 3
**Presentation:** 4
**Contribution:** 3
**Rating:** 6
**Confidence:** 4

**Summary:**

This paper studies the one-shot isolated sign language recognition problem. The method is to learn sign embeddings with sign classification pretraining and then adapt to new vocabularies with nearest neighbors classification. The method works relatively well, and the most influential factor for generalization is pretraining dataset size+diversity. The tool is released for adaptation by the community.

**Strengths:**

* I like the 1-shot ISLR task, and it is well motivated (both socially and technically).
* Reasonable method/experiments. Empirical confirmation of key hypotheses (pose-based methods abstract away much irrelevant information, sign diversity helps generalization).
* The artifact of the paper was released as a usable tool.

**Weaknesses:**

* Could use more references to related work. The PopSign dataset paper itself (https://neurips.cc/virtual/2023/poster/73414) should be cited when talking about its Kaggle competition. Project Shuwa (https://github.com/google/shuwa) has a similar approach using knn, though it is not described academically. SignCLIP (https://arxiv.org/abs/2407.01264) is slightly borderline timewise since it came out on July 1, but it is a more general embedding approach that is evaluated on few-shot ISLR (but performs much worse without finetuning). You should better describe how your approach differs from De Coster & Dambre (2023), besides pretraining on ASL Citizen.
* There are some additional experiments you could do with relatively little effort given the existing setup, but I'm not holding this against you too much since there could always be additional experiments. But for example, measuring how performance scales with additional shots beyond 1 doesn't require any additional training but could inform data collection. (I know this might not be possible for datasets with only 1 shot though.)
* It would have been nice to see more qualitative error analysis beyond what's in lines 470-481.
* There is the claim throughout the paper that "Isolated sign language recognition (ISLR) is a crucial first step toward achieving full sign language translation (SLT).", but I don't see evidence for this, especially given the strength of "crucial". Perhaps "promising" or "potential". The version on lines 462-464 is a little better since it's talking specifically about futureproofing, but I don't see how the experiments support the claim that operating on individual signs is "crucial", vs. e.g. training on new data reflective of language change. It feels a bit weird to make negative claims about other sign language translation methods not working when the paper doesn't engage with any sign language translation literature. The claim on 516-519 seems much more justified.

**Questions:**

No particular questions, except clarification on the nit above about "crucial".

I expect to increase the score to 8 with improvements to related work.

---

> ### Author Response · Authors · 2024-11-26
>
> We thank you for your constructive feedback and for providing the references to PopSign, Shuwa, SignCLIP, and other related works. Your suggestions for improving the paper's clarity will be implemented in future revisions.

---

### Official Review · Reviewer_t1Dw · 2024-11-01

**Soundness:** 2
**Presentation:** 1
**Contribution:** 1
**Rating:** 3
**Confidence:** 4

**Summary:**

The paper proposes to formulate isolated sign language recogniton (ISLR) as a one-shot classification task. The method consists of a pose-based SLR model and a dictionary. During inference, the prediction can be obatined by retreiving the dictionary. The method achieves competitive performance on two benchmarks, WLASL and AUTSL.

**Strengths:**

1. It is reasonable to formulate ISLR as a few(one)-shot problem.
2. The model is pose-based which is inherently efficient.
3. The method generalizes well across different sign languages.

**Weaknesses:**

1. The techinical contribution is limited. Although the formulation makes sense, the method is too simple to provide new insights to the community.
2. There is a significant lack of comparison with existing methods. In Tab. 3, the authors only report the performance of their own method. The authors should compare with more SOTA methods.
3. The presentation should be improved. There are a lot of black boxes in the figures, which are not quite informative. The authors are suggested to use different colours to highlight important model architectures.
4. It is good to develop a real-world applications for DHH. Conducting a user study for the app would improve the paper.

**Questions:**

n/a

---

> ### Author Response · Authors · 2024-11-26
>
> Thank you for your comments. While our method is straightforward, its effectiveness under specific pre-training conditions highlights the importance of dataset selection in sign language recognition. We believe that the findings of this paper are novel (for the sign language recognition field) and that the method's limited novelty is outweighed by this. We will consider submitting a revised version to a more application-oriented venue.

---

### Official Review · Reviewer_g2HC · 2024-11-01

**Soundness:** 2
**Presentation:** 1
**Contribution:** 2
**Rating:** 3
**Confidence:** 4

**Summary:**

This paper introduces a one-shot approach for isolated sign language recognition (ISLR). The approach involves two main steps. First, a classification network is pre-trained on a multilingual ISLR dataset to learn an embedding space that captures visual cues. Second, a support dictionary, containing one entry per class, is used to compute the embedding for each class. During inference, the classification probabilities for each test video are calculated by applying softmax to the similarity scores between the test video’s embedding and each support entry's embedding.

The proposed method can also be applied when the embedding model is pre-trained on different sign languages, as various sign languages share a common visual modality. Experiments demonstrate that this method is relatively robust to variations in the one-shot support set and the selection of the pre-training sign language. As the first one-shot ISLR method, it has few direct baselines for comparison. It achieves a one-shot MRR of 50.8% on a 10,235-entry dictionary set, significantly surpassing random selection. Additionally, the authors have developed an application to enable collaboration with the DHH (Deaf and Hard of Hearing) community. However, due to the double-blind review policy, details and visualizations of this application are limited.

**Strengths:**

1. The paper is the first one-shot approach for ISLR. One-shot classification has significant real-world applications in isolated sign language recognition, especially as large-scale corpora are unavailable for many sign languages while one-shot dictionaries are much easier to collect.
2. The pretrained embedding space enables one-shot ISLR in _any_ sign language, creating potential for broad impact within the global deaf community.

**Weaknesses:**

1. While this paper addresses a meaningful, practical problem, its methodological approach lacks novelty. The adopted one-shot learning method has been widely used in various areas of computer vision for nearly a decade, to the point that it has become almost conventional.

Isolated sign language recognition is fundamentally an image or video classification task, and similar one-shot methods have been utilized extensively in prior work, including:
* Vinyals, Oriol, et al. Matching networks for one-shot learning. Advances in Neural Information Processing Systems, 29 (2016).
* Tian, Yonglong, et al. Rethinking few-shot image classification: A good embedding is all you need? Proceedings of the 16th European Conference on Computer Vision, ECCV 2020. Springer International Publishing, 2020.

The proposed one-shot method is also analogous to linear probe classification commonly used in self-supervised representation learning, as seen in works such as:
* Chen, Ting, et al. A simple framework for contrastive learning of visual representations. International Conference on Machine Learning. PMLR, 2020.
* Grill, Jean-Bastien, et al. Bootstrap your own latent: A new approach to self-supervised learning. Advances in Neural Information Processing Systems, 33 (2020): 21271-21284.
* Oquab, Maxime, et al. Dinov2: Learning robust visual features without supervision. arXiv preprint arXiv:2304.07193 (2023).

Although one-shot learning has not been directly applied to isolated sign language recognition (ISLR), the concept of classifying new instances based on a reference dictionary has been explored in other tasks, such as sign spotting in continuous sign language videos, which is even more complex than ISLR.
* Varol, Gül, et al. Scaling up sign spotting through sign language dictionaries. International Journal of Computer Vision, 130.6 (2022): 1416-1439.

It should be noted that the referenced literature is not exhaustive. The proposed one-shot method shares a fundamentally similar approach with this extensive body of prior work. Although the paper introduces this idea to a new task, it lacks novelty in other technical aspects, including training strategies and model architecture.

2. The writing and figures in this paper require large refinement. First, more detailed information about the proposed method is needed, including the classifier’s architecture and the algorithmic procedure. Currently, the authors provide only an extremely brief overview between lines 147 and 152, with a reference to Mittal et al. (2021). However, this reference primarily discusses an attention mechanism, which does not clarify the one-shot method being used. Second, the orders of different sections need re-organization. Currently, the datasets and evaluation metrics are included in 'Methodology' but it is more proper to describe them in 'Experiments' and use some abstract symbols to denote datasets in the method description. Finally, Figures 1 and 4 are fairly underdeveloped and fail to convey essential information effectively.

**Questions:**

1. The reviewer requests a more detailed description of the algorithm's procedure, specifically including the steps for pretraining and one-shot inference, which are currently missing from the submission.

2. The reviewer would like the authors to provide arguments that highlight the novelty of their approach compared to conventional one-shot methods used in other image classification tasks.

3. The reviewer would appreciate additional details on the developed application, specifically how it relates to the one-shot learning problem and its potential benefits for the deaf community.

---

> ### Author Response · Authors · 2024-11-26
>
> We appreciate your valuable feedback and the suggested references. The proposed comparison with linear probe classification will strengthen our paper's related works section. Your structural suggestions are also appreciated and will be considered for improvement.

---

### Official Review · Reviewer_BVXz · 2024-11-05

**Soundness:** 3
**Presentation:** 3
**Contribution:** 1
**Rating:** 5
**Confidence:** 4

**Summary:**

This paper introduces a one-shot learning approach for isolated sign language recognition, allowing for language-independent and scalable recognition of signs without extensive retraining. The method includes pre-training a model to create embeddings for signs, as well as using a dense vector search to match new signs, supporting rapid recognition of large, evolving vocabularies. Developed with feedback from the Deaf and Hard of Hearing (DHH) community, the goal of the proposed framework is to offer adaptability across multiple sign languages, ensuring it can be improved over time based on community needs.

**Strengths:**

+ The paper is clearly-written.

+ The proposed method deals with sign language recognition, which is a challenging problem with important social impact.

+ The paper includes a very nice overview and motivation of the problem. It also includes a very convincing discussion regarding the importance of one-shot learning.

+ It also describes a strategy of co-creation with the DHH community, in terms of the methodology and goals. This is important, since the actual end-users of such technologies can provide concrete feedback that can positively affect the effectiveness of the developed pipelines.

**Weaknesses:**

- The proposed framework for one-shot inference has relatively limited novelty. There are several well-established methods that leverage embedding spaces and attention mechanisms to compare a query instance against a support set of examples, in a very similar manner, for both images and temporal data. See for example:

Vinyals, O., Blundell, C., Lillicrap, T. and Wierstra, D., 2016. Matching networks for one shot learning. Advances in neural information processing systems, 29.

Snell, J., Swersky, K. and Zemel, R., 2017. Prototypical networks for few-shot learning. Advances in neural information processing systems, 30.

Sung, F., Yang, Y., Zhang, L., Xiang, T., Torr, P.H. and Hospedales, T.M., 2018. Learning to compare: Relation network for few-shot learning. In Proceedings of the IEEE conference on computer vision and pattern recognition (pp. 1199-1208).

Zhu, L. and Yang, Y., 2018. Compound memory networks for few-shot video classification. In Proceedings of the European conference on computer vision (ECCV) (pp. 751-766).

(Bahdanau et al., 2014)

(Mittal et al. 2021)


- The experimental evaluation is insufficient. The proposed method is only compared with I3D in Section 4.1. The authors' argument for that is that in recent bibliography, this is reported as the best performing SOTA method. However, comparisons with additional SOTA methods are crucial, in order to have a more clear understanding on the pros and cons of the proposed method over the previous ones. Furthermore, in the experimental protocol of the paper, there might be other SOTA methods that perform better than I3D, which is an additional reason why more SOTA methods should have been included in the comparisons.


- In addition, the experimental evaluation of Section 4.2 ("One-shot SLR") includes only internal comparisons of the proposed method. It does not include any quantitative evaluation with any SOTA methods at all. This is highly inadequate.

**Questions:**

In the rebuttal, I'd very interested in reading the authors' arguments on all my points of criticism.

---

> ### Author Response · Authors · 2024-11-26
>
> Thank you for your insightful comments and for sharing relevant references. We acknowledge the need for further methodological refinement and will incorporate your suggestions in future revisions.

---

### Note · Authors · 2024-11-26

I have read and agree with the venue's withdrawal policy on behalf of myself and my co-authors.